# Association between Maternal Origin, Pre-Pregnancy Body Mass Index and Caesarean Section: A Nation-Wide Registry Study

**DOI:** 10.3390/ijerph18115938

**Published:** 2021-06-01

**Authors:** Fatou Jatta, Johanne Sundby, Siri Vangen, Benedikte Victoria Lindskog, Ingvil Krarup Sørbye, Katrine Mari Owe

**Affiliations:** 1Institute of Health and Society, Department of Community Medicine and Global Health, University of Oslo, 0317 Oslo, Norway; fatou.jatta@studmed.uio.no (F.J.); johanne.sundby@medisin.uio.no (J.S.); 2Norwegian Research Centre for Women’s Health, Department of Obstetrics and Gynecology, Oslo University Hospital, 0424 Oslo, Norway; sirvan@ous-hf.no (S.V.); isorbye@ous-hf.no (I.K.S.); 3Institute of Clinical Medicine, University of Oslo, 0318 Oslo, Norway; 4Section for Diversity Studies, Oslo Metropolitan University, 0130 Oslo, Norway; benedik@oslomet.no; 5Department of Child health and development, Norwegian Institute of Public Health, 0213 Oslo, Norway

**Keywords:** caesarean section, delivery mode, maternal origin, maternal birthplace, pre-pregnancy BMI, migrant women, length of residence, registry based

## Abstract

Aims: To explore the association between maternal origin and birthplace, and caesarean section (CS) by pre-pregnancy body mass index (BMI) and length of residence. Methods: We linked records from 118,459 primiparous women in the Medical Birth Registry of Norway between 2013 and 2017 with data from the National Population Register. We categorized pre-pregnancy BMI (kg/m^2^) into underweight (<18.5), normal weight (18.5–24.9) and overweight/obese (≥25). Multinomial regression analysis estimated crude and adjusted relative risk ratios (RRR) with 95% confidence intervals (CI) for emergency and elective CS. Results: Compared to normal weight women from Norway, women from Sub-Saharan Africa and Southeast Asia/Pacific had a decreased risk of elective CS (aRRR = 0.57, 95% CI 0.37–0.87 and aRRR = 0.56, 0.41–0.77, respectively). Overweight/obese women from Europe/Central Asia had the highest risk of elective CS (aRRR = 1.42, 1.09–1.86). Both normal weight and overweight/obese Sub-Saharan African women had the highest risks of emergency CS (aRRR = 2.61, 2.28-2.99; 2.18, 1.81-2.63, respectively). Compared to women from high-income countries, the risk of elective CS was increasing with a longer length of residence among European/Central Asian women. Newly arrived migrants from Sub-Saharan Africa had the highest risk of emergency CS. Conclusion: Women from Sub-Saharan Africa had more than two times the risk of emergency CS compared to women originating from Norway, regardless of pre-pregnancy BMI.

## 1. Introduction

Caesarean section (CS) is a surgical procedure for delivery of the fetus, and it is increasing in both high-, middle- and low-income countries [1]. Although CS may be medically indicated and a lifesaving intervention to safeguard maternal or infant health [2], it is associated with an increased risk of maternal morbidity and mortality compared to vaginal deliveries, leading to both short-term and long-term complications for the mother [3,4]. Studies have also shown that newborns delivered by CS have a higher risk of respiratory problems, admission to a neonatal intensive care unit and perinatal mortality [5].

The CS rate varies between countries but also within countries between migrant and non-migrant women [6]. Non-medical factors, such as fear of birth, maternal age, network support, lack of health insurance, communication barriers, education, literacy, and lifestyle behavior, may partly explain differences in CS rates [7,8,9]. Some studies have reported higher rates of CS among migrant women compared to women born in the host country [8], but studies are inconclusive as both a higher and a lower rate of CS has been observed [10]. 

Emergency CS is, in contrast to elective CS, associated with an increased risk of maternal morbidity and mortality, including postpartum hemorrhage and severe sepsis, and may create emotional as well as psychological distress for the woman [1,11]. Some studies have reported that emergency CS is more prevalent in women from Latin America & the Caribbean compared to women from Sub-Saharan Africa [12,13]. Others have reported that migrant women of African descent have an increased risk of emergency CS [14]. 

The prevalence of overweight and obesity among women of reproductive age is in-creasing in many countries [15,16], with some reporting higher prevalence among pregnant migrant women compared to pregnant non-migrant women [17,18,19]. A high pre-pregnancy body mass index (BMI) is a known risk factor for CS [20,21], especially for the emergency type [22]. Overweight or obese women who have a CS experience more adverse outcomes when compared to normal weight women with a CS or obese women with a vaginal delivery [23]. Migrant women from countries with a lower prevalence of overweight or obesity compared to the host country may adapt the norm of the host country after a longer residence [24]. 

Even though disparities in maternal outcomes between migrants and non-migrants are well documented [25,26], few studies have explored the association between maternal origin and subtypes of CS (elective and emergency type) in relation to pre-pregnancy BMI. Our primary aim was therefore to explore the association between maternal region of origin and birthplace, with elective and emergency CS by pre-pregnancy BMI in primiparous women. Secondly, we estimated whether the association changed across different lengths of residence.

## 2. Materials and Methods

### 2.1. Study Design and Participants

This was a prospective nationwide registry-based cohort study linking records from 608,980 deliveries recorded in the Medical Birth Registry of Norway (MBRN) with data from the National Population Register (NPR) between 2008 and 2017. We excluded 349,865 multiparous women to avoid the influence of having a cesarean section (CS) in a previous pregnancy. Women giving birth to newborns with a birth weight less than 500 g (*n* = 935) or with a pregnancy length less than 20 weeks or more than 43 weeks (*n* = 2554) were excluded to avoid errors in gestational length estimates of births at borderline of viability and postdates (Figure 1). We also excluded multiple gestations (*n* = 8709), placenta abruptio (*n* = 719), placenta previa (*n* = 622) and fetal presentation incompatible with vaginal birth (*n* = 613). Furthermore, women with missing data on maternal country of birth (*n* = 1988) and foreign-born women with two Norwegian-born parents (*n* = 2209) were omitted from the study. Due to high proportions of missing values on pre-pregnancy BMI, we limited our study period to 2013 and 2017. Finally, the study sample included 118,459 primiparous women.

### 2.2. Outcome

The main outcome variables were elective and emergency cesarean section (CS) as registered in the MBRN. A CS is defined as an emergency CS when the decision to perform the CS is made less than 8 h before delivery. When the decision to perform the CS is made more than 8 h before delivery, it is defined as an elective CS. We re-categorized unspecified CS (*n* = 67) as emergency CS. The quality of CS subtypes in the MBRN is good [27].

### 2.3. Exposures

The exposure of interest was maternal origin, including both maternal region of origin and maternal birthplace. Maternal region of origin is defined as the women’s ethnic origin, determined by the birthplace of her mother. Maternal birthplace is defined as where the woman was born and further categorized into foreign-born and Norwegian-born. Women who were born abroad to two foreign-born parents or foreign-born with one Norwegian-born parent were classified as foreign-born. We classified women born in Norway with two, one or no Norwegian-born parents as Norwegian-born. We used the global burden of disease (GBD) framework [28], which is based on epidemiological similarity and geographic closeness, to categorize maternal country of origin and birthplace into one of the following regions: Norway, High-income countries, Europe & Central Asia, Sub-Saharan Africa, North Africa & the Middle East, South Asia, Southeast Asia, East Asia & the Pacific, Latin America & the Caribbean. The exposure variables were obtained from the National Population Register (NPR).

Maternal weight and height are registered at the first antenatal care visit, which usually takes place in pregnancy weeks 8–12. Maternal pre-pregnancy BMI was calculated using weight in kilograms divided by the square of height in meters (kg/m^2^). We categorized BMI using the WHO classification for underweight (<18.5 kg/m^2^), normal weight (18.5–24.9 kg/m^2^; reference category), overweight (25.0–29.9 kg/m^2^), obesity class I (30.0–34.9 kg/m^2^) and obesity class II (≥35 kg/m^2^). We merged overweight and obesity class I and II into one category (≥25.0 kg/m^2^). We observed high proportions of missing values on pre-pregnancy BMI due to the gradual registration in the MBRN from 2006 onwards (*n* = 27,087, 22.9%). We also observed large variations in the registration of pre-pregnancy BMI between health regions. The missing values of BMI were randomly distributed between the different categories of BMI, with risk of bias less likely. Hence, we included women with missing values on pre-pregnancy BMI as a separate category.

Information on length of residence was obtained from NPR and categorized into three groups (0–4 years, ≥5 years and <0 years). Length of residence was calculated as the time interval between the year of first immigration to Norway and the year of delivery. Newly arrived migrants were those with a length of residence 0–4 years. Women giving birth in Norway before lawful immigration were registered with a negative length of residence. The latter group included a high proportion of women in our study (*n* = 4909, 15.0%) due to increasing case processing times for new arrivals in Norway after 2008, which has caused a delay in the registration of length of residence in the registry. Most women in this group migrated to Norway due to family reunion (57.5%) or work (26.0%). 

### 2.4. Study Factors

We included maternal age (in years) both as a continuous and categorical variable (≤19–24, 25–34 and ≥35 years). Maternal education is registered in the National education database (Statistics Norway) as the highest completed level of education (in years) and categorized as no education/primary (≤10 years), secondary (11–14 years), university/college (14–20 years). We dichotomized marital status as married/not married. Smokers were defined as either smoking before or during pregnancy. We also included infant’s year of birth, Norwegian health regions (southeast, west, mid and north) and paternal birthplace (Norwegian-born or foreign-born) as covariates.

### 2.5. Statistical Analysis

The proportions of demographic variables by mode of delivery were compared by cross-tabulations. We used a multinomial regression analysis (using vaginal delivery as the reference) to simultaneously estimate the associations between maternal origin, and elective and emergency CS as the relative risk ratio (RRR) with 95% confidence intervals (CI). Norwegian women comprised the reference group in the main analysis. We used stratified analysis to explore if the associations with elective and emergency CS differed by categories of pre-pregnancy BMI. Models were adjusted for maternal age, education, marital status, infant’s year of birth, smoking and paternal birthplace. Due to the wide variations in the registration of pre-pregnancy BMI and varied CS rates between Norwegian health regions during the study period, we also adjusted for health regions. Furthermore, we stratified the analyses on length of residence in a separate analysis restricted to foreign-born women, using women from high-income countries as the reference. When assessing the association between maternal region of origin and elective and emergency CS in strata of length of residence, we adjusted for pre-pregnancy BMI. 

Additionally, we performed two sensitivity analyses with elective and emergency CS as the outcome. First, we removed those with gestational diabetes, pre-existing diabetes, pregnancy-induced hypertension or preeclampsia (*n* = 11,718), as these conditions were considered risk factors for or associates of caesarean delivery. Secondly, we excluded women who gave birth to a large-for-gestational age (LGA, above the 90th percentile) or small-for-gestational-age (SGA, below the 5th percentile) newborn (*n* = 26,461) to determine if these exclusions changed the estimates because these women may be at increased risk of having a CS. We ran both sensitivity analyses with and without information on pre-pregnancy BMI. Thirdly, we performed a complete case analysis including only women with information on pre-pregnancy BMI, excluding 27,087 women (22.9%). We performed this sensitivity analysis to explore the possible influence of pre-pregnancy BMI on the association between maternal region of origin and CS in a complete case setting. 

All statistical analyses were performed using IBM SPSS Statistics version 26 (IBM Corp., Inc., Armonk, NY, USA) and Stata IC version 16 (Stata Statistical Software, College Station, TX, USA). 

## 3. Results

### 3.1. Maternal Characteristics

During the study period between 2013 and 2017, 118,495 deliveries were recorded among primiparous women in Norway, of which 3.2% were delivered by elective and 13.8% by emergency caesarean section (CS) (Table 1). Both elective and emergency CS varied by maternal age, pre-pregnancy BMI, marital status, education, health regions and paternal birthplace. 

The highest proportions of elective CS were observed among women from Latin America & the Caribbean (3.7%), and Europe & Central Asia (3.7%) compared to 3.3% among women from Norway. One-fourth of the women from Sub-Saharan Africa and one-fifth of the Latin American & Caribbean women had an emergency CS (25.3% and 22.1%), whereas the corresponding proportion among Norwegian women was 12.9%.

Emergency CS was more prevalent among women with pre-pregnancy overweight or obesity (18.7%) than among women with normal weight (11.9%). We also observed higher proportions of both elective and emergency CS among women who developed preeclampsia or gestational diabetes mellitus (GDM) and among those delivering an SGA or an LGA newborn. 

### 3.2. Delivery Mode by Maternal Region of Origin

Rates of elective and emergency CS varied among women from the top three regions of origin between 2013 and 2017, but showed less variability among Norwegian women during the same period of time (Figure 2). 

We also observed large variations in the proportions of elective and emergency CS by maternal regions of origin stratified on pre-pregnancy BMI (Appendix A). Albeit few underweight women in our study (3.8%), both elective and emergency CS were common among underweight women from Latin America & the Caribbean (7.5% and 17.5%). Normal and overweight or obese women from Sub-Saharan Africa had the highest proportions of emergency CS (24.3% and 33.6%). Norwegian women with normal weight or overweight/obesity had the lowest proportions of emergency CS (10.8% and 17.6%). 

### 3.3. Maternal Region of Origin and Caesarean Section by Pre-Pregnancy BMI

#### 3.3.1. Elective CS

Compared to underweight women from Norway, underweight women from most regions had a decreased risk of elective CS except women from Latin America & the Caribbean (aRRR = 2.28, 95% CI 0.66–7.83) (Table 2). None of these associations reached statistical significance due to too few cases. Normal weight women from Sub-Saharan Africa and Southeast Asia, East Asia & the Pacific had the lowest risks of elective CS (aRRR = 0.57, 95% CI 0.37-0.87; aRRR = 0.56, 95% CI 0.41–0.77, respectively). Overweight and obese women from Europe & Central Asia had highest risk of elective CS compared to Norwegian women with overweight and obesity (aRRR = 1.42, 95% CI 1.09–1.86). 

#### 3.3.2. Emergency CS

Among the underweight women, we observed the highest risk of emergency CS among women from South Asia (aRRR = 2.59, 95% CI 1.45–4.62). Compared to normal weight women from Norway, women from the other regions except the high-income countries had increased risks of emergency CS, with the highest risk observed among Sub-Saharan African women (aRRR = 2.61, 95% CI 2.28–2.99). We observed the highest risk of emergency CS among overweight and obese Sub-Saharan African women (RRR = 2.18, 95% CI 1.81–2.63).

### 3.4. Maternal Region of Origin and Caesarean Section by Pre-Pregnancy BMI among Foreign Born Women

In a separate analysis limited to foreign-born women (*n* = 32,688), we observed similar associations with both elective and emergency CS but with overall higher risk estimates compared to women from high-income countries (Table 3). Women from Sub-Saharan Africa had more than 2.5 times increased risk of emergency CS across all categories of pre-pregnancy BMI. Underweight women from Latin America & the Caribbean had more than three times increased risk of emergency CS compared to underweight women from high-income countries. Few underweight women had an elective CS, and we did not observe significant associations in this group.

### 3.5. Maternal Region of Origin and Elective and Emergency CS by Length of Residence 

Half of the foreign-born women (17,213 out of 32,688) had a short length of residence (0–5 years) and 32.3% had a long length of residence (≥5 years). Overall, women with a long length of residence had the highest rates of both elective and emergency CS (4.1% and 17.5%). Table 4 shows the crude and adjusted association between maternal region of origin and elective and emergency CS by length of residence. Compared to women from high-income countries, women from Europe & Central Asia with a long length of residence (≥5 years) had an increased risk of elective CS (aRRR = 1.90; 95% CI 1.36–2.66). Sub-Saharan African women had the highest risk of emergency CS in all strata of length of residence with the highest risk observed among newly arrived migrants (aRRR = 2.70, 95% CI 2.19–3.32). We also observed increased risks of emergency CS in women with both a long and short length of residence from South Asia, Southeast Asia, East Asia & the Pacific, and Latin America & the Caribbean (Table 4). 

### 3.6. Sensitivity Analysis

Results from the sensitivity analysis did not change the strength or direction of the risk estimates from the main analysis.

## 4. Discussion

### 4.1. Main Findings 

We observed that primiparous women from Sub-Saharan Africa had more than two times the risk of emergency CS across each category of pre-pregnancy BMI compared to women originating from Norway. Among underweight women from South Asia, the risk of emergency CS was more than 2.5 times the risk in underweight women from Norway. Although not statistically significant, underweight women from Latin America & the Caribbean also had more than two times the risk of emergency CS compared to underweight women from Norway. Overweight and obese women from Europe & Central Asia had an increased risk of elective CS compared to overweight and obese women originating from Norway. Conversely, we observed a decreased risk of elective CS among normal weight women originating from regions of Sub-Saharan Africa and Southeast Asia, East Asia & the Pacific compared to normal weight Norwegian women. 

Among foreign-born women, we observed higher risk estimates for both elective and emergency CS compared to women from high-income countries. Underweight women from Latin America & the Caribbean had more than three times the risk of emergency CS than underweight women from high-income countries. 

We observed an increased risk of emergency CS among most foreign-born women in categories of lengths of residence compared to women from high-income countries. Migrant women from Europe & Central Asia with a long length of residence had an increased risk of elective CS compared to women from high-income countries. The highest risk of emergency CS was observed among newly arrived migrant women from Sub-Saharan Africa compared to women from high-income countries. 

### 4.2. Strengths and Limitations

The major strengths of our study are the prospective study design with a large sample size, linkage to high-quality national registries and a validated outcome. We were also able to study the association with subtypes of caesarean delivery. The richness of data with comprehensive information on demographic characteristics, pregnancy- and migration-related factors allowed us to consider a wide range of confounding factors. Using a unique personal identification number also made exploration of maternal region of origin, length of residence and paternal birthplace possible.

Few underweight women had an elective CS in our study. Consequently, we observed weak associations between maternal origin and elective CS among the underweight women. This was even more evident in the analysis limited to foreign-born women only, among whom we observed only 47 elective caesareans. Hence, these results need to be interpreted with caution. 

The high proportion of missing information on pre-pregnancy BMI is a limitation to our study. However, missing information on BMI were at random due to gradual introduction in the national birth registry, and the risk of bias is therefore less likely to have influenced our results. We also performed a sensitivity analysis limited to women with information on pre-pregnancy BMI without changing the risk estimates. 

Previous studies have shown that language barriers are associated with a higher risk of caesarean section [29,30]. We did not have information on language proficiency. To take into account the impact of acculturation, we stratified analysis by length of residence. 

Even though we adjusted for a wide range of possible confounding factors, residual or unmeasured confounding factors could have influenced the estimated associations in our study. 

### 4.3. Interpretation

While previous studies have estimated the association between maternal origin and elective and emergency caesarean section in different populations [12,31,32], few other studies have examined how pre-pregnancy BMI influences the association among migrant women. Normal weight women have been shown to have a lower risk of both elective and emergency CS compared to women with overweight and obesity [33]. In our study, normal weight women from Sub-Saharan Africa and South Asia had a decreased risk of elective CS compared to normal weight women from Norway. The risk of emergency CS in women originating from these regions, on the other hand, was high. Furthermore, we observed the highest risk of emergency CS among normal weight women. For elective CS, the risk was only slightly elevated in overweight or obese women from Europe & Central Asia. Underweight women from Latin America & the Caribbean had more than two times the risk of elective CS compared to underweight women from Norway. Even though the absolute number of elective CS was low, this is an interesting finding because these women come from a region where CS is frequently performed. The results from this study suggest that differences in pre-pregnancy BMI have less influence on the association between maternal origin and subtypes of CS than previously reported [34].

Even though the rate of CS in Sub-Saharan Africa is low, migrant women from this region have an increased risk of CS when giving birth in a host country. The observation that Sub-Saharan African women were at high risk of emergency CS compared to Norwegian women is in line with previous studies [35,36]. Most of the Sub-Saharan women in our study came from Somalia, Eritrea and Ethiopia where female genital mutilation (FGM) is highly prevalent. We did have information on FGM. However, a recently published study from Norway showed that the high risk of CS in Somali women was not related to FGM [37].

Our study observed that many migrant women had an increased risk of emergency CS, regardless of length of residence, compared to women from high-income countries. Adjusting for pre-pregnancy BMI did not change the estimates substantially. Other studies support this finding [8,38,39]. Only women with a long length of residence from Europe & Central Asia had elevated risk of elective CS. Most of the women in this group came from Poland and Lithuania, in which CS rates are higher than in Norway. The response to pain during childbirth may vary among women from different regions of origin and may thereby affect mode of delivery due to negative perceptions toward having a vaginal delivery [40]. Maternal preference and fear of childbirth has also been reported to be a contributing factor to the increased rates of elective CS among migrant women [41,42]. We observed variations in elective and emergency CS among women originating from African and Asian regions as well as Latin America & the Caribbean, but there seemed to be a steady pattern among Norwegian women. However, the study period in our study was only five years, and a longer time may have showed different patterns.

Studies have shown that non-medical factors, such as language barriers, support during labor and birth, distinct attitudes to caesarean section as a mode of delivery among the woman herself and her family, lack of knowledge and misconceptions of the procedure and views on the management of deliveries, may partly explain the differences of CS among migrant women [41,43,44]. Insufficient knowledge of health may prevent these women from taking part in the decision making concerning their own health. One limitation of our study was that we had no data to assess health literacy among these women. 

The cultural and social beliefs and their complex interaction may presumably explain why many migrant women are not attending antenatal care and present late during labor. Inadequate antenatal care or the inability to communicate their health problems may lead to undiagnosed and untreated pregnancy complications among migrant women [43]. Consequently, poorer pregnancy and birth outcomes are frequently reported among migrant women, particularly among women of African origin [41]. A common cultural belief among some groups of migrant women is that they should not be touched by caregivers due to fear of miscarriage. Preference of a female caregiver is also reported [44]. We did not have information on social or cultural beliefs or on how these complex interactions may have influenced the delivery of antenatal care and childbirth. Hence, we can only speculate about their explanatory values. 

Language barriers may hinder good communication about symptoms, advice and medical explanations, occasionally resulting in cesarean section [7,45]. Migrant women from Sub-Saharan Africa with both a short and long length of residence had an increased risk of emergency CS with the highest risk among those with a short length of residence. Conversely, women from Europe & Central Asia with a long length of residence had an increased risk of elective CS compared to women from high-income countries. Even though we did not observe a reduced risk of CS with a longer residence, the importance of language proficiency for birth outcome could still be in play. Stereotyping against a group has been indicated among healthcare professionals [46], resulting in unequal care to migrant women, particularly during stressful situations such as delivery [47].

Other factors such as low socioeconomic status and education may also be important risk factors for cesarean delivery. However, the risk estimates did not change substantially when adjusting for a wide range of confounding factors, including education. In contrast to a Swedish study [32], including paternal birthplace in the analysis did not change the risk estimates. Differences in the prevalence of pregnancy complications, such as gestational diabetes and preeclampsia, may also partly explain the differences in emergency CS observed in our study. In a sensitivity analysis restricted to women without these complications, the associations between maternal region of origin with elective and emergency type remained unchanged. Even though the proportions of LGA and SGA differed with maternal origin, in a separate analysis excluding women who gave birth to an LGA or SGA newborn, the risk of elective and emergency CS was not altered. 

## 5. Conclusions

Few studies have examined the relationship between maternal origin and cesarean section separately for elective and emergency type, elucidating the importance of pre-pregnancy BMI and length of residence. We observed a strong association with emergency CS among Sub-Saharan African women in all categories of pre-pregnancy BMI. Language barriers and late detection of women with medical indications for a CS may explain the low rate of elective CS and the high rate of emergency CS among these women. Although newly arrived migrants from Sub-Saharan Africa had the highest risk of emergency CS, the risk was more than twice the risk in high-income women independent of their length of residence, while women from Latin America & the Caribbean had the highest risk of elective CS regardless of length of residence.

### Implications for Practice

Sub-Saharan African women are at higher risk of emergency CS, which also causes the most complications among these women regardless of their pre-pregnancy BMI or duration of residence in Norway. Given that most of the CS performed among these women may not be medically indicated, this is a pointer to health care providers to pay attention to this group of women during pregnancy and childbirth to reduce emergency CS.

## Figures and Tables

**Figure 1 ijerph-18-05938-f001:**
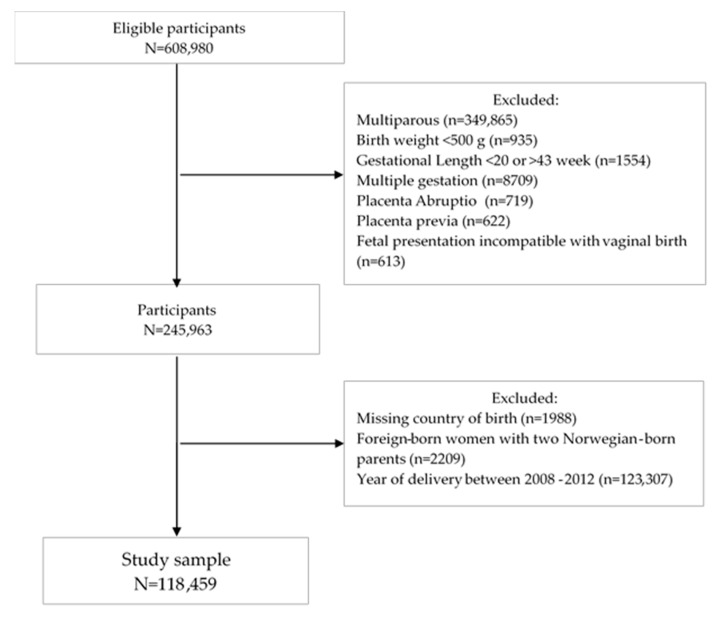
Flow chart of the study population.

**Figure 2 ijerph-18-05938-f002:**
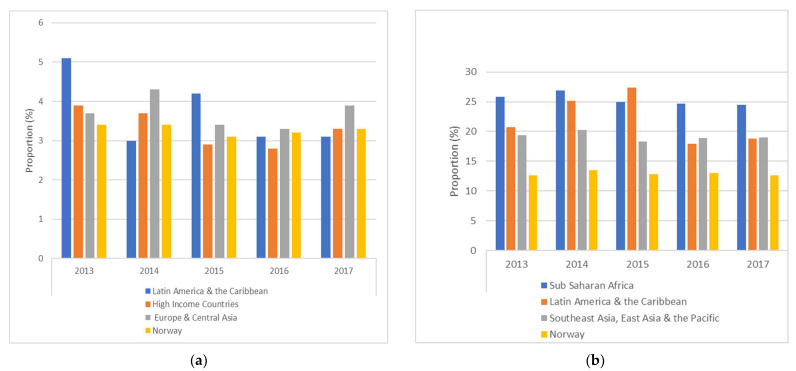
Proportions of elective (**a**) and emergency (**b**) caesarean section by the top 3 maternal regions of origin between 2013 and 2017.

**Table 1 ijerph-18-05938-t001:** Maternal characteristics by mode of delivery among primiparous women, *n* = 118,459.

Characteristics		Vaginal Delivery *n* = 98,247 (82.9%)	Elective CS *n* = 3816 (3.2%)	Emergency CS *n* = 16,396 (13.8%)
**Maternal region of origin (GBD)**				
Norway		62,067 (83.8)	2414 (3.3)	9562 (12.9)
High-income countries		12,338 (83.5)	489 (3.3)	1949 (13.2)
Europe Central Asia		10,787 (83.6)	476 (3.7)	1642 (12.7)
Sub-Saharan Africa		2835 (72.4)	89 (2.3)	992 (25.3)
North Africa & the Middle East		3770 (81.9)	142 (3.1)	690 (14.9)
South Asia		1827 (80.2)	58 (2.6)	392 (17.2)
Southeast Asia, East Asia & the Pacific		3847 (78.6)	109 (2.2)	938 (19.2)
Latin America & the Caribbean		776 (74.2)	39 (3.7)	231 (22.1)
**Maternal age:**	≤24 years	23,385 (88.8)	524 (2.8)	2712 (9.3)
	25–34 years	65,778 (82.7)	2436 (3.2)	10,881 (14.1)
	≥35 years	9084 (67.5)	856 (8.4)	2803 (24.1)
**Pre-pregnancy BMI (kg/m^2^):**	Underweight (<18.5)	3922 (87.7)	143 (3.2)	407 (9.1))
	Normal weight (18.5–24.9)	50,676 (85.1)	1744 (2.9)	7122 (11.9)
	Overweight/Obese (≥25)	21,304 (77.9)	939 (3.4)	5115 (18.7)
	*Missing Information*	*22,345 (82.5)*	*990 (3.7)*	*3752 (13.9)*
**Married**		90 173 (83.1)	3428 (3.2)	14,876 (13.7)
**Educational level ^1^**	No education/Primary	13,646 (81.5)	536 (3.2)	2564 (15.3)
	Secondary	21,283 (82.4)	855 (3.3)	3693 (14.3)
	University/College	54,915 (83.7)	2128 (3.2)	8558 (13.1)
	Unknown	8403 (81.7)	297 (2.9)	1581 (15.4)
**Smokers**		8497 (80.8)	385 (3.7)	1634 (15.5)
**SGA 5th percentile**		5396 (75.2)	272 (3.8)	1508 (21.0)
**LGA 10th percentile**		4509 (68.9)	318 (4.9)	1715 (26.2)
**Preeclampsia**		2666 (60.6)	133 (3.0)	1599 (36.4)
**GDM**		3826 (70.3)	267 (4.9)	1352 (24.8)
**Norwegian Health regions:**	Southeast West Mid North	55,956 (82.1)	2545 (3.7)	9661 (14.2)
		21,900 (86.2)	543 (2.1)	2954 (11.6)
		12,646 (81.1)	453 (2.9)	2369 (15.3)
		7745 (82.1)	275 (2.9)	1412 (14.9)
**Paternal birthplace: ^1^**	Foreign-born	70,004 (83.4)	2720 (3.2)	11,224 (13.4)
	Norwegian-born	25,288 (82.5)	943 (3.1)	4436 (14.5)

^1^ Missing information paternal birthplace (*n* = 3844).

**Table 2 ijerph-18-05938-t002:** Association between maternal region of origin and elective and emergency CS by pre-pregnancy BMI (kg/m^2^), *n* = 118,459.

	Elective CS	Underweight (BMI < 18.5) *n* = 143 (3.2%)		Normal (BMI 18.5–24.9) *n* = 1744 (2.9%)		Overweight/obese (BMI ≥ 25.0) *n*= 939 (3.4%)		Missing BMI *n* = 990 (3.7%)	
**Maternal Region of Origin**	**Cases** ***n* (%)**	**RRR (95% CI)**		**RRR (95% CI)**		**RRR (95% CI)**		**RRR (95% CI)**	
		**Unadjusted**	**Adjusted ***	**Unadjusted**	**Adjusted ***	**Unadjusted**	**Adjusted ***	**Unadjusted**	**Adjusted ***
Norway	2414 (3.3)	Reference	Reference	Reference	Reference	Reference	Reference	Reference	Reference
High-income countries	489 (3.3)	0.85 (0.49–1.49)	0.79 (0.44–1.41)	1.06 (0.92–1.23)	0.99 (0.85–1.15)	0.88 (0.71–1.10)	0.77 (0.61–0.98)	1.11 (0.92–1.34)	1.03 (0.81–131)
Europe & Central Asia	476 (3.7)	0.84 (0.53–1.33)	0.94 (0.53–1.68)	1.16 (1.01–1.33)	1.17 (0.98–1.40)	1.47 (1.18–1.83)	1.42 (1.09–1.86)	1.04 (0.85–1.29)	1.17 (0.86–1.58)
Sub-Saharan Africa	89 (2.3)	0.82 (0.41–1.65)	0.86 (0.37–1.96)	0.70 (0.49–1.00)	0.57 (0.37–0.87)	1.30 (0.87–1.92)	1.23 (0.79–1.91)	0.63 (0.41–0.96)	0.75 (0.41–1.35)
North Africa & the Middle East	142 (3.1)	0.54 (0.22–1.35)	0.68 (0.25–1.82)	1.12 (0.87–1.43)	1.23 (0.93–1.62)	0.78 (0.52–1.17)	0.64 (0.39–1.05)	0.97 (0.71–1.35)	1.54 (0.99–2.38)
South Asia	58 (2.6)	0.22 (0.03–1.60)	0.27 (0.04–2.04)	0.87 (0.51–1.28)	0.86 (0.56–1.33)	1.31 (0.81–2.12)	1.37 (0.82–2.27)	0.55 (0.31–0.99)	0.64 (0.28–1.48)
Southeast Asia, East Asia & Pacific	109 (2.2)	0.31 (0.12–0.77)	0.32 (0.13–0.82)	0.68 (0.51–0.90)	0.56 (0.41–0.77)	0.83 (0.50–1.38)	0.74 (0.43–1.25)	0.99 (0.71–1.40)	0.92 (0.60–1.41)
Latin America & the Caribbean	39 (3.7)	2.31 (0.69–7.74)	2.28 (0.66–7.83)	1.49 (0.96–2.32)	1.01 (0.61–1.66)	0.91 (0.37–2.23)	0.65 (0.24–1.79)	1.10 (0.58–2.09)	1.08 (0.48–2.36)
	**Emergency CS**	**Underweight (BMI < 18.5)** ***n*** **= 407 (9.1%)**		**Normal (BMI 18.5–24.9)** ***n* = 7122 (12.0%)**		**Overweight/obese (BMI ≥ 25.0)** ***n* = 5115 (18.7%)**		**Missing BMI** ***n* = 3752 (13.9%)**	
**Maternal Region of Origin**	**Cases** ***n* (%)**	**RRR (95% CI)**		**RRR (95% CI)**		**RRR (95% CI)**		**RRR (95% CI)**	
		**Unadjusted**	**Adjusted ***	**Unadjusted**	**Adjusted ***	**Unadjusted**	**Adjusted ***	**Unadjusted**	**Adjusted ***
Norway	9562 (12.9)	Reference	Reference	Reference	Reference	Reference	Reference	Reference	Reference
High-income countries	1949 (13.2)	0.98 (0.67–1.45)	0.86 (0.57–1.30)	1.04 (0.96–1.12)	0.97 (0.90–1.06)	1.10 (0.99–1.21)	1.03 (0.93–1.14)	1.03 (0.92–1.15)	1.03 (0.92–1.15)
Europe & Central Asia	1642 (12.7)	1.03 (0.75–1.41)	1.01 (0.68–1.49)	1.13 (1.04–1.22)	1.11 (1.01–1.22)	1.04 (0.92–1.18)	0.97 (0.84–1.12)	0.96 (0.85–1.09)	0.96 (0.85–1.09)
Sub-Saharan Africa	992 (25.3)	2.03 (1.46–2.97)	2.06 (1.31–3.22)	2.63 (2.35–2.95)	2.61 (2.28–2.99)	2.40 (2.05–2.81)	2.18 (1.81–2.63)	2.13 (1.84–2.47)	2.13 (1.84–2.47)
North Africa & the Middle East	690 (14.9)	1.44 (0.93–2.23)	1.39 (0.81–2.37)	1.24 (1.09–1.41)	1.25 (1.07–1.45)	1.27 (1.09–1.49)	1.28 (1.06–1.54)	1.14 (0.96–1.35)	1.14 (0.96–1.35)
South Asia	392 (17.2)	2.24 (1.36–3.72)	2.59 (1.45–4.62)	1.32 (1.11–1.58)	1.48 (1.23–1.79)	1.66 (1.34–2.05)	1.64 (1.29–2.08)	1.38 (1.11–1.71)	1.38 (1.11–1.71)
Southeast Asia, East Asia & Pacific	938 (19.2)	1.83 (1.33–2.52)	1.52 (1.06–2.18)	1.86 (1.68–2.06)	1.65 (1.47–1.84)	1.56 (1.29–1.889	1.37 (1.12–1.68)	1.67 (1.43–1.95)	1.67 (1.43–1.95)
Latin America & the Caribbean	231 (22.1)	2.75 (1.19–6.36)	2.37 (0.99–5.65)	2.02 (1.63–2.51)	1.62 (1.29–2.04)	1.67 (1.19–2.33)	1.42 (0.99–2.04)	2.30 (1.75–3.01)	2.30 (1.75–3.01)

* Adjusted for maternal age, infant’s year of birth, education, marital status, smoking, paternal birthplace and Norwegian health regions. RRR = relative risk ratio. 95% CI = 95% confidence intervals.

**Table 3 ijerph-18-05938-t003:** Association between maternal region of origin and elective and emergency CS by pre-pregnancy BMI (kg/m^2^) among foreign-born women (*n* = 32,688).

	Elective CS	Underweight (BMI < 18.5) *n* = (%)		Normal (BMI 18.5–24.9) *n* = (%)		Overweight/obese (BMI ≥ 25.0) *n* = (%)		Missing BMI *n* = (%)	
**Maternal region of origin**	**Cases** **%**	**RRR (95% CI)**		**RRR (95% CI)**		**RRR (95% CI)**		**RRR (95% CI)**	
		**Unadjusted**	**Adjusted ***	**Unadjusted**	**Adjusted ***	**Unadjusted**	**Adjusted ***	**Unadjusted**	**Adjusted ***
High-income countries	3.6	Reference	Reference	Reference	Reference	Reference	Reference	Reference	Reference
Europe & Central Asia	3.6	0.92 (0.37–2.31)	1.50 (0.56–3.99)	0.94 (0.75–1.17)	1.18 (0.92–1.51)	1.53 (1.05–2.24)	2.19 (1.43–3.35)	0.85 (0.62–1.17)	1.24 (0.80.1.91)
Sub-Saharan Africa	2.3	0.90 (0.31–2.65)	1.76 (0.51–6.07)	0.55 (0.36–0.84)	0.63 (0.38–1.06)	1.36 (0.81–2.29)	1.96 (1.05–3.63)	0.61 (0.37–0.99)	1.26 (0.62–2.57)
North Africa & Middle East	3.1	0.59 (0.17–2.14)	1.31 (0.34–5.06)	0.92 (0.68–1.26)	1.32 (0.93–1.87)	0.83 (0.49–1.43)	1.10 (0.59–2.06)	0.88 80.58–1.33)	2.03 (1.14–3.59)
South Asia	2.3	^1^	^1^	0.61 (0.35–1.05)	0.70 (0.38–1.29)	1.19 (0.59–2.42)	1.91 (0.90–4.05)	0.56 (0.29–1.11)	0.67 (0.23–1.95)
Southeast Asia, East Asia & Pacific	2.5	0.42 (0.13–1.39)	0.52 (0.15–1.76)	0.62 (0.44–0–87)	0.59 (0.41–0.85)	1.22 (0.67–2.20)	1.27 (0.68–2.39)	0.96 (0.63–1.45)	1.16 (0.68–1.98)
Latin America & Caribbean	4.1	3.95 (0.92–16.99)	4.23 (0.95–18.91)	1.28 (0.78–2.11)	0.94 (0.54–1.64)	0.99 (0.35–2.82)	0.78 (0.23–2.62)	1.15 (0.58–2.27)	1.51 (0.65–3.52)
	**Emergency CS**	**Underweight (BMI < 18.5)** ***n* = (%)**		**Normal (BMI 18.5–24.9)** ***n* = (%)**		**Overweight/obese (BMI≥ 25.0)** ***n* = (%)**		**Missing BMI** ***n* = (%)**	
**Maternal Region of Origin**	**Cases** **%**	**RRR (95% CI)**		**RRR (95% CI)**		**RRR (95% CI)**		**RRR (95% CI)**	
		**Unadjusted**	**Adjusted ***	**Unadjusted**	**Adjusted ***	**Unadjusted**	**Adjusted ***	**Unadjusted**	**Adjusted ***
High-income countries	14.2	Reference	Reference	Reference	Reference	Reference	Reference	Reference	Reference
Europe & Central Asia	12.8	0.88 (0.50–1.55)	1.14 (0.61–2.13)	0.95 (0.84–1.08)	1.09 (0.95–1.25)	0.84 (0.70–1.01)	0.94 (0.77–1.14)	0.95 (0.78–1.14)	1.07 (0.84–1.36)
Sub-Saharan Africa	26.6	1.75 (0.97–3.17)	2.98 (1.49–5.99)	2.35 (2.02–2.74)	2.89 (2.41–3.47)	2.15 (1.75–2.65)	2.49 (1.93–3.22)	2.22 (1.81–2.73)	2.73 (2.03–3.67)
North Africa & Middle East	15.8	1.31 (0.68–2.52)	1.93 (0.92–4.05)	1.11 (0.9–1.31)	1.32 (1.09–1.59)	1.08 (0.87–1.34)	1.32 (1.03–1.69)	1.20 (0.96–1.51)	1.51 (1.11–2.05)
South Asia	17.7	1.48 (0.64–3.40)	2.55 (1.05–6.19)	1.19 (0.94–1.50)	1.55 (1.21–1.98)	1.34 (0.99–1.80)	1.57 (1.13–2.18)	1.42 (1.07–1.89)	1.70 (1.16–2.49)
Southeast Asia, East Asia & Pacific	20.9	1.59 (0.89–2.83)	1.82 (0.97–3.40)	1.76 (1.52–2.03)	1.78 (1.52–2.07)	1.51 (1.18–1.93)	1.44 (1.11–1.87)	1.83 (1.48–2.26)	1.70 (1.31–2.22)
Latin America & Caribbean	23.9	2.79 (0.99–7.86)	3.30 (1.11–9.72)	1.86 (1.46–2.38)	1.69 (1.30–2.19)	1.54 (1.04–2.28)	1.43 (0.94–2.16)	2.48 (1.80–3.40)	2.28 (1.52–3.42)

* Adjusted for maternal age, infant’s year of delivery, education, marital status, smoking, paternal birthplace and Norwegian health regions. RRR = relative risk ratio. 95% CI = 95% confidence intervals. ^1^ Due to a small number of underweight South Asian women having an elective CS (*n* = 35/1503).

**Table 4 ijerph-18-05938-t004:** Association between maternal region of origin and elective and emergency CS by length of residence among foreign-born women (*n* = 32,688).

Length of Residency	Elective CS					
	0–4 years *n* = 486 (46.2%)		≥5 years *n* = 428 (40.7%)		<0 years *n* = 138 (13.1%)	
	RRR (95% CI)		RRR (95% CI)		RRR (95% CI)	
	Unadjusted	Adjusted *	Unadjusted	Adjusted *	Unadjusted	Adjusted *
**Maternal region of origin**						
High-income countries	Reference	Reference	Reference	Reference	Reference	Reference
Europe & Central Asia	0.93 (0.70–1.22)	1.07 (0.79–1.44)	1.58 (1.17–2.13)	1.90 (1.36–2.66)	0.74 (0.43–1.26)	0.66 (0.37–1.16)
Sub-Saharan Africa	0.76 (0.50–1.15)	0.74 (0.44–1.25)	0.80 (0.48–1.35)	1.02 (0.56–1.84)	0.82 (0.35–1.87)	0.95 (0.39–2.31)
North Africa & the Middle East	0.70 (0.47–1.07)	0.78 (0.49–1.24)	1.15 (0.77–1.72)	1.56 (0.99–2.46)	0.77 (0.39–1.51)	0.70 (0.33–1.48)
South Asia	0.47 (0.24–0.92)	0.51 (0.25–1.04)	1.37 (0.75–2.52)	1.84 (0.95–3.56)	0.40 (0.09–1.74)	0.20 (0.03–1.55)
Southeast Asia, East Asia & Pacific	0.52 (0.33–0.82)	0.48 (0.47–1.82)	1.04 (0.69–1.57)	1.27 (0.82–1.96)	0.46 (0.17–1.23)	0.35 (0.12–1.07)
Latin America & the Caribbean	1.14 (0.61–2.13)	0.92 (0.47–1.82)	1.44 (0.68–3.05)	1.44 (0.65 (3.23)	1.76 (0.69–4.50)	1.16 (0.38–3.57)
**Length of residency**	**Emergency CS**					
	**0–4 years** ***n* = 2815 (52.2%)**		**≥5 years** ***n* = 1849 (34.3%)**		**<0 years** ***n* = 727 (13.5%)**	
	**RRR (95% CI)**		**RRR (95% CI)**		**RRR (95% CI)**	
	**Unadjusted**	**Adjusted ***	**Unadjusted**	**Adjusted ***	**Unadjusted**	**Adjusted ***
**Maternal region of origin**						
High-income countries	Reference	Reference	Reference	Reference	Reference	Reference
Europe & Central Asia	0.92 (0.79–1.06)	1.01(0.86–1.18)	1.07 (0.91–1.26)	1.13 (0.95–1.35)	0.68 (0.50–0.91)	0.71 (0.51–0.97)
Sub-Saharan Africa	2.37 (2.00–2.80)	2.70 (2.19–3.32)	2.12 (1.74–2.59)	2.36 (1.88–2.97)	1.91 (1.33–2.73)	1.93 (1.26–2.95)
North Africa & the Middle East	1.14 (0.95–1.38)	1.23 (0.99–1.52)	1.21 (0.99–1.48)	1.26 (0.99–1.58)	0.82 (0.57–1.18)	0.98 (0.66–1.46)
South Asia	1.20 (0.95–1.53)	1.39 (1.07–1.81)	1.33 (0.97–1.83)	1.60 (1.14–2.24)	1.37 (0.84–2.23)	1.57 (0.93–2.68)
Southeast Asia, East Asia & Pacific	1.70 (1.43–2.01)	1.72 (1.43–2.06)	1.71 (1.42–2.05)	1.85 (1.52–2.25)	1.47 (1.01–2.13)	1.58 (1.06–2.36)
Latin America & the Caribbean	1.86 (1.40–2.46)	1.71 (1.27–2.31)	1.62 (1.11–2.35)	1.64 (1.11–2.43)	1.92 (1.15–3.21)	2.02 (1.17–3.48)

* Adjusted for pre-pregnancy BMI, infant’s year of birth, education, marital status, smoking, paternal birthplace and Norwegian health regions.

## Data Availability

The data presented in this study are available on request from the corresponding author. The data are not publicly available due to the registry design of the study.

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
