# Peer review of "Association between Maternal Origin, Pre-Pregnancy Body Mass Index and Caesarean Section: A Nation-Wide Registry Study"

_ijerph, 2021, doi:10.3390/ijerph18115938_

Round 1
Reviewer 1 Report
This study is a medical records review to examine the rates of Cesarean Section in women in Norway stratified by the maternal birthplace and maternal body mass. The authors find that women from Latin America and Sub-Saharan Africa had the highest rates of CS. In the conclusion, this is attributed to communication barriers including language and education.
Minor comments:
There are many places in the manuscript where multiple citations are not combined correctly (e.g line 46: “[7,8] [9]” along with blank lines (eg. Line 449: “70. “) in the bibliography. This is likely the result of broken links from external citation software and therefore all the citations should be thoroughly verified to ensure they reference the intended sources.
Line 85-86 is awkwardly phrased (or missing a word) and difficult to understand.
Line 94: replace missing reference.
The low resolution of Figure 1 makes it difficult to read.
There is a typo in the figure 2b legend “South EastAsia” should be Southeast Asia and “EastAsia” should have a space.
Reviewer 2 Report
This is a relatively large-scale prospective study that examines elective and emergency c-section rates in Norway with evaluation for pre-pregnancy BMI, maternal origin, birthplace, and length of residence in N Norway.
The study was carefully designed and took into account previous work germane to the author's research. The article is well written, nicely organized, and cites current literature that is central to the work of the study. To my surprise and delight, each question that I raised in my review of the paper was answered by the authors in the body of the manuscript.
My only suggestion is that the authors make explicit in the discussion that social and cultural factors are inherently complex and that they offer some (even if brief) comment on how those factors might play out in the delivery of prenatal care and delivery. Likely there are both educational and biological factors that are woven into cultural constructs around childbirth and prenatal care. I recognize that the authors cannot make hard claims about the nature of these complex interactions, but by making some explicit reference to social and cultural elements, it may help readers to be cognizant of hidden variables that cannot realistically be included in such a study as this.
Excellent work.
